# Expression Profile and Diagnostic Significance of MicroRNAs in Papillary Thyroid Cancer

**DOI:** 10.3390/cancers14112679

**Published:** 2022-05-28

**Authors:** Mariusz Rogucki, Iwona Sidorkiewicz, Magdalena Niemira, Janusz Bogdan Dzięcioł, Angelika Buczyńska, Agnieszka Adamska, Katarzyna Siewko, Maria Kościuszko, Katarzyna Maliszewska, Anna Wójcicka, Jakub Supronik, Małgorzata Szelachowska, Joanna Reszeć, Adam Jacek Krętowski, Anna Popławska-Kita

**Affiliations:** 1Department of Endocrinology, Diabetology and Internal Medicine, Medical University of Białystok, 15-276 Białystok, Poland; mariusz.rogucki@sd.umb.edu.pl (M.R.); ak001@wp.pl (A.A.); katarzynasiewko@o2.pl (K.S.); maria.kosciuszko@umb.edu.pl (M.K.); katarzyna.maliszewska2@umb.edu.pl (K.M.); jakubsupronik@uskwb.pl (J.S.); mszelachowska@poczta.onet.pl (M.S.); adam-kretowski@wp.pl (A.J.K.); 2Clinical Research Centre, Medical University of Białystok, 15-276 Białystok, Poland; iwona.sidorkiewicz@umb.edu.pl (I.S.); magdalena.niemira@umb.edu.pl (M.N.); angelika.buczynska@umb.edu.pl (A.B.); 3Department of Human Anatomy, Medical University of Białystok, 15-089 Białystok, Poland; janusz.dzieciol@umb.edu.pl; 4Warsaw Genomics Inc., 01-682 Warszawa, Poland; anna.wojcicka@warsawgenomics.pl; 5Department of Medical Pathomorphology, Faculty of Medicine, Medical University of Białystok, 15-269 Białystok, Poland; joasia@umb.edu.pl

**Keywords:** microRNA, papillary thyroid cancer, biomarker

## Abstract

**Simple Summary:**

In recent years, the incidence of papillary thyroid cancer (PTC) has increased in many countries worldwide. MicroRNAs appear to be important regulators of PTC, but a better understanding of their role is needed to develop novel diagnostic tools and identify potential vulnerabilities. In this study, we aimed to gain insight into the microRNA profile of PTC tissue. Consequently, crucial pathways in PTC were highlighted. A panel of four microRNAs (miR-152-3p, miR-221-3p, miR-551b-3p, and miR-7-5p) was proposed as a PTC diagnostic biomarker. Our analysis indicated that microRNAs are a potential diagnostic tool for PTC patients.

**Abstract:**

The incidence of papillary thyroid cancer (PTC) has increased in recent years. To improve the diagnostic management of PTC, we propose the use of microRNAs (miRNAs) as a biomarker. Our aim in this study was to evaluate the miRNA expression pattern in PTC using NanoString technology. We identified ten miRNAs deregulated in PTC compared with reference tissue: miR-146b-5p, miR-221-3p, miR-221-5p, miR-34-5p, miR-551b-3p, miR-152-3p, miR-15a-5p, miR-31-5p, and miR-7-5p (FDR < 0.05; |fold change (FC)| ≥ 1.5). The gene ontology (GO) analysis of differentially expressed miRNA (DEM) target genes identified the predominant involvement of epidermal growth factor receptor (EGFR), tyrosine kinase inhibitor resistance, and pathways in cancer in PTC. The highest area under the receiver operating characteristic (ROC) curve (AUC) for DEMs was found for miR-146-5p (AUC = 0.770) expression, indicating possible clinical applicability in PTC diagnosis. The combination of four miRNAs (miR-152-3p, miR-221-3p, miR-551b-3p, and miR-7-5p) showed an AUC of 0.841. Validation by real-time quantitative polymerase chain reactions (qRT-PCRs) confirmed our findings. The introduction of an miRNA diagnostic panel based on the results of our study may help to improve therapeutic decision making for questionable cases. The use of miRNAs as biomarkers of PTC may become an aspect of personalized medicine.

## 1. Introduction

Thyroid cancer remains the most common malignancy among endocrine tumors. In 2020, an estimated 586,000 cases of thyroid cancer were diagnosed worldwide [1]. In addition, the estimated number of papillary thyroid cancer (PTC) cases in 2022 in the United States was 43,800 [2]. PTC accounts for the majority of all differentiated thyroid cancer diagnoses [3] and is the most common type of thyroid malignancy, with its incidence notably increasing in recent decades [4]. The increase in the rate of diagnosed PTC cases is primarily due to detection improvements, even in small, indolent thyroid nodules [5]. Despite having a good prognosis, up to 10% of identified PTC cases metastasize or contribute to death during cancer progression [6]. Nevertheless, the prognosis for the majority of PTCs is favorable in terms of long-term survival, although recurrence occurs in 25–30% of PTC patients [7,8,9].

Ultrasound, commonly used in screening PTC, has detected thyroid nodules in up to 68% of examined patients [10]. However, the gold standard for detecting PTC is fine-needle aspiration biopsy (FNAB) performed under ultrasound guidance [11]. FNAB is inexpensive, highly accurate, minimally invasive, and safe, with a low complication rate [12]. However, up to 10% of the obtained FNAB results are nondiagnostic [13]. In addition, FNAB does not provide complete information about the clinical nature of tumors, such as the presence of angioinvasion features, which may affect further management [14]. As such, novel diagnostic biomarkers that are equivalent thyroid cancer diagnostic tools need to be identified [15,16].

Numerous molecular alterations, such as changes in microRNA (miRNA) expression, are associated with the clinicopathological features of PTCs [17]. Moreover, miRNAs are implicated in many tumor-promoting gene deregulations in PTC [18,19]. Several miRNAs, their effect on PTC cells, and the potential utility as therapeutic targets and/or prognostic markers have been studied [20]. Nevertheless, there is still a need to identify the pathways that could be possibly targeted and be considered useful for facilitating PTC diagnosis. Considering PTC is a heterogeneous neoplasm from both morphological and molecular points of view, comprehensive research on epigenetic alterations as potential diagnostic markers in PTC is of high importance [21]. Introducing miRNA expression patterns to the diagnostic protocol may also aid personalized clinical management. Undoubtedly, miRNA diagnostic applications may improve PTC detection, thus reducing the need for patients to undergo invasive procedures. Therefore, our broad goal in the current study was to evaluate miRNA expression patterns in early PTC compared to normal thyroid gland tissue. Consequently, the results of this study highlight the new molecular features of PTC and provide insight into the events underlying its development and progression. Moreover, we assessed the diagnostic significance of selected miRNAs as possible PTC biomarkers.

## 2. Materials and Methods

### 2.1. Study Subjects

We conducted this study using postoperative material from the Department of Endocrinology Diabetology and Internal Medicine at the Medical University of Bialystok in Poland. We collected PTC tissues from patients with different tumor stages according to tumor-node-metastasis (TNM) classification and confirmed by a histopathological examination. We obtained the specimens from postoperative formalin-fixed paraffin-embedded (FFPE) blocks. The control group consisted of specimens from the same patients with normal nontumor tissues. Control tissues were confirmed by a pathomorphologist and did not contain PTC cells or other lesions. Due to rapid tumor progression, we could not obtain nontumor tissues from 2 patients with PTC. Patients without a diagnosis of PTC upon postoperative histopathological examination and patients with metastatic PTC in the lymph nodes were excluded from the study. Additional exclusion criteria included other comorbidities, chronic conditions, immunosuppressive treatment, and cigarette smoking. Consequently, we included 80 FFPE tissue samples (41 PTC tissues and 39 reference tissues without malignant processes) in the study (Table 1). The study protocol was approved in advance by the Bioethics Committee of the Medical University of Białystok (approval number: R-I-002/491/2019), and written informed consent was obtained from each participant.

### 2.2. Detection of the miRNA Profiles

We extracted miRNAs from 3 consecutive 10 μm-thick FFPE sections. A pathologist identified tumor-rich regions (at least 50% of the tissue area evaluated was PTC cells) of interest to minimize the risk of false-negative results. The total RNA, including the miRNA fraction, was isolated using a deparaffinization solution and an miRNeasy FFPE extraction kit according to the manufacturer’s protocol (Qiagen, Hilden, Germany). We assessed the miRNA concentration using a Qubit (Invitrogen, Carlsbad, CA, USA). We prepared the samples for nCounter miRNA expression profiling according to the manufacturer’s recommendations (NanoString Technologies, Seattle, WA, USA). Briefly, we prepared 100 ng of miRNA samples by ligating a specific miR tag onto the 3′ end of each mature miRNA, followed by an overnight hybridization (65 °C) with nCounter Reporter and Capture probes. Subsequently, the samples were placed into the nCounter Prep Station for automated sample purification and reporter capture. Each sample was scanned on an nCounter Digital Analyzer for data collection. We deposited the NanoString data in the Gene Expression Omnibus (GEO) database (GSE191117).

### 2.3. Validation of the NanoString Results by Real-Time Quantitative Polymerase Chain Reaction (qRT-PCR)

For the validation experiments, we used a miRCURY LNA RT Kit (Qiagen), according to the manufacturer’s protocol, to reverse transcribe the miRNA samples (41 PTC samples and 39 control thyroid tissues) on a BioRad S1000 thermal cycler (Mississauga, ON, Canada). Then, the candidate miRNAs were quantified by qRT-PCR using specific primers and a miRCURY LNA SYBR Green PCR Kit (Qiagen). We ran the samples in duplicate on a LightCycler 480 Real-Time PCR System (Roche, Basel, Switzerland). We used miR-103a-3p and U6 snRNA as the endogenous controls, and the NormFinder algorithm (Department of Molecular Medicine (MOMA), Aarhus University Hospital, Aarhus N, Denmark) was used to calculate their stability. The relative miRNA expression was calculated using qBase MSExcel VBA based on multiple samples and multiple reference miRNAs. the qBase converts the Ct values from all runs within one experiment to normalized and rescaled quantities that can be visualized in graphs [22]. Fold change (FC) values were provided using the GeneGlobe Data Analysis Center (Qiagen; geneglobe.qiagen.com, accessed on 23 February 2022).

### 2.4. Data Analysis

We used the nSolver 4.0 Analysis software (NanoString) for data analysis and normalization using the average geometric mean of the top 100 probes detected. We used a false discovery rate (FDR) correction for multiple comparisons, limited to 0.5, to adjust the *p* values. The threshold value for the significance used to define differentially expressed miRNAs (DEMs) was |fold change (FC)| of ≥1.5.

Statistical analyses were performed with GraphPad PRISM (v. 9.1.1.; GraphPad Software, San Diego, CA, USA). Preliminary statistical analysis (Shapiro–Wilk test) revealed that the studied parameters did not follow a normal distribution. Consequently, we performed nonparametric tests between groups. All the data are presented as medians and ranges. The areas under the receiver operating characteristic (ROC) curves (AUCs) for DEMs were calculated.

We prepared a logistic regression model by combining the DEMs, and we used the WEKA tool (Weka, version 3.8.5, Hamilton, New Zealand) to discriminate between PTC and the control tissue. For the comparison of PTC vs. control, we chose specific miRNAs for the feature selection. Models based on logistic regression, naive Bayes, and tree-based J48 algorithms were established. For each combination, confusion matrices were prepared to evaluate the model. We measured the performance of the multi-miRNA classifiers using the classification precision and the AUC.

### 2.5. miRNA Target Prediction and Functional Annotation of the Selected miRNA Targets

To examine the functions of the identified miRNAs, miRNA target prediction was performed using MicroRNA ENrichment TURned NETwork (MIENTURNET, Rome, Italy) [23]. We used the following online databases and carried out Gene Ontology (GO) analysis and functional annotation clustering to examine the functions of the identified target genes: the Gene Ontology enrichment analysis and visualization tool (GOrilla; http://cbl-gorilla.cs.technion.ac.il/, accessed on 2 February 2022), DAVID (GO and Kyoto Encyclopedia of Genes and Genomes (KEGG), Enrichment Analysis: https://david.ncifcrf.gov/, accessed on 2 February 2022, g: Profiler (https://biit.cs.ut.ee/gprofiler/gos, accessed on 2 February 2022), and Metascape (https://metascape.org, accessed on 2 February 2022). To identify highly connected hub genes in protein–protein interactions, we used the Search Tool for Retrieval of Interacting Genes/Proteins (STRING) database (with an interaction score > 0.4) [24] and the CytoHubba plugin based on Cytoscape version 3.8.2 (http://cytoscape.org/, accessed on 30 March 2022) [25].

## 3. Results

### 3.1. miRNA Profile in Papillary Thyroid Cancer

To determine the tumor-specific miRNA expression pattern, we profiled the tissue expression of 798 miRNAs using the NanoString Technology platform. The differential expression of ten miRNAs was detected between the studied groups (Table 2). Specifically, eight miRNAs were upregulated and two were downregulated in PTC compared to controls based on a threshold FDR < 0.05 and an |FC| ≥ 1.5.

### 3.2. miRNA Target Genes

We predicted the DEMs’ target genes using MIENTURNET, a web tool designed to obtain computationally predicted and/or experimentally validated target genes from TargetScan and miRTarBase. The results indicated 725 putative target genes for the miRNAs based on a minimum threshold of two miRNA–target interactions. The STRING database was used to visualize the protein relationships (Figure 1).

### 3.3. Functional Enrichment Analysis

To obtain insights into the biomolecular significance of the identified target genes of the DEMs, we performed GO analysis with various databases and obtained enriched GO terms. The GO analysis of the PTC network biological process revealed a predominant role of the following categories: ‘nervous system development’, ‘neurogenesis’, and ‘regulation of nitrogen compound metabolic process’. ‘Nucleoplasm’, ‘nuclear lumen’, and ‘intracellular organelle’ were among the significantly associated cellular component categories. Furthermore, the genes of the GO molecular functions are involved in ‘enzyme binding’, ‘protein kinase binding’, and ‘protein binding’. Additionally, the Kyoto Encyclopedia of Genes and Genomes (KEGG) pathway enrichment analysis confirmed that ‘epidermal growth factor receptor (EGFR) tyrosine kinase inhibitor resistance’, ‘pathways in cancer’, and ‘mammalian target of rapamycin (mTOR) signaling pathway’ are predominantly involved in PTC’s pathogenesis (Figure 2).

### 3.4. Hub Gene Identification

To further examine the protein–protein interactions (PPIs) of the DEM target genes, we used the Cytoscape software; we used the Maximal Clique Centrality (MCC) algorithm in the cytoHubba plugin to identify the top 10 hub genes in the PPI network (Figure 3, Table 3). The top 10 hub genes were mitogen-activated protein kinase 1 (*MAP2K1*), Raf-1 proto-oncogene, serine/threonine kinase (*RAF1*), insulin receptor substrate 1 (*IRS1*), epidermal growth factor receptor (*EGFR*), KIT proto-oncogene receptor tyrosine kinase (*KIT*), kinase insert domain receptor (*KDR*), Erb-B2 receptor tyrosine kinase 3 (*ERBB3*), NRAS proto-oncogene, GTPase (*NRAS*), phosphoinositide-3-kinase regulatory subunit 1 (*PIK3R1*), and phosphatidylinositol-4,5-bisphosphate 3-kinase catalytic subunit beta (*PIK3CB*).

### 3.5. Receiver Operating Characteristic Curve Analysis

We evaluated the diagnostic value of the DEMs as candidate PTC biomarkers using the AUCs. All the identified DEMs showed significantly higher AUCs compared with AUC = 0.500, excluding miR-31-5p (*p* > 0.05). The highest AUC, indicating possible clinical usefulness in PTC diagnosis, was observed for miR-146-5p (AUC = 0.770), miR-551-3p (AUC = 0.740), and miR-222-3p (AUC = 0.720) (Figure 4).

### 3.6. Logistic Regression Model

Using a logistic regression model, we screened the DEMs for constructing an miRNA-based PTC diagnostic signature. Logistic regression is a supervised learning algorithm commonly used for binary classification tasks. Intercept and coefficients are parts of the model, intercept is the log odds when the variable is 0, this means the absence of PTC. Coefficients describe how much the log odds change with the presence of PTC. True positive, false positive, precision, and AUC values describe the diagnostic usefulness of the model.

The feature selection chose four DEMs with the highest diagnostic values, and a logistic regression model was constructed. The parameters of the model and the common quality measures are summarized in Table 4. The AUC value obtained for a combination of miR-152-3p + miR-221-3p + miR-551b-3p + miR-7-5p (AUC = 0.841) had a higher diagnostic value than the highest AUC for an miRNA used separately—miR-146-5p (AUC = 0.770).

### 3.7. Data Validation

We validated the DEMs from the miRNA-based diagnostic signature to verify the results of the NanoString analysis. We investigated 80 FFPE samples from the PTC and control groups using qRT-PCR to determine the relative expression levels of the four discovered DEMs (Figure 5A–D). The validation results demonstrated high similarity to the expression profiles established by NanoString, thus confirming the importance of all the DEMs included in the logistic regression model.

## 4. Discussion

The rapid development of high-resolution technologies has increased the detection rate for thyroid cancer [26]. However, the invasive FNAB procedure still remains the diagnostic tool of choice [27]. Despite the high accuracy, safety, and cost effectiveness of this method [28], patients undergoing FNAB are at risk of several complications, such as a hematoma at the injection site or pain via ecchymosis, swelling, and inadvertent punctures to the trachea, carotid artery, or jugular vein [29]. Moreover, up to 10% of FNAB results are nondiagnostic [30]. Hence, novel screening and diagnostic markers, using highly specialized techniques that can increase the PTC detection rate, are still being evaluated. Our results confirmed that FFPE tissues are suitable resources for such miRNA expression analyses and could be useful in the management algorithm of patients with thyroid nodules.

The further identification of potential biomarkers useful in the diagnosis and prognosis of PTC is needed. In our study, we determined the overall miRNA expression profile with NanoString technology and the RT–PCR method was used to confirm selected miRNAs expression. MiRNAs can easily be detected in various types of specimens by methods such as microarrays, nCounter technology, or RT-PCR. The large-scale methods allow the analysis of miRNA patterns that comprise numerous miRNAs providing valuable insight into the regulation of many biological processes. Considering miRNA analysis in PTC, the method used (small RNA sequencing, microarray, PCR method), type of specimen (FFPE, FNAB, frozen tissue), inclusion/exclusion criteria (presence of comorbidities, metastases), type of reference tissue (normal tissue paired from cancer patients, normal tissue from healthy patients, benign tissue), and histologic type of cancer should also be taken into consideration when comparing and interpreting the results [31,32]. Reports on selective miRNA expression using the PCR method do not allow to assess the miRNA fingerprint for diagnosis. Thus, the improved standardization of methods used to evaluate miRNAs expression may support their introduction in the personalized medicine approach in PTC patients. Significant literature data have already been accumulated on the particular miRNA expression in a specific type of specimens in PTC [33,34,35,36], however, miRNA expression pattern using large-scale methods with subsequent data validation are still needed to fill in the data gap. Our miRNA combination distinguishes between early PTC and normal tissue, thus additional studies including benign tissues should be performed to confirm the diagnostic usefulness of the miRNA panel. What should be also noted is that the proposed diagnostic tool is incapable of differentiating metastatic PTC from normal thyroid gland tissue. Metastatic PTC were excluded to obtain homogenous PTC cohort, which was also helpful to comprehensively describe tumor-initiating alterations in miRNA expression pattern [37]. Literature data suggest distinct miRNA expression pattern associated with lymph node metastasis in PTC patients [38]. It is worth noting that the data on the diagnostic value of the miRNA profile in differentiation between metastatic PTC and non-metastatic PTC patients are limited.

Notably, miRNA deregulation has been implicated in the development of many different cancers, including thyroid cancer [20,39]. Literature data show a relationship between the expression of various miRNAs in thyroid cancer and the TNM classification [17,36]. The evaluation of PTC pathogenesis particularly emphasizes the deregulation of miR-146b, miR-221, miR-222, miR-181b, and miR-21 [40]. Deregulation of miR-146b affects the MAPK/ERK and TGF-β pathways, thus increasing the risk of vascular invasion and metastasis to lymph nodes and distant organs [36]. Another miRNA implicated in the pathogenesis of PTC is miR-181b whose upregulation affects the inhibition of apoptosis and promotion of cell division [41]. Increased miR-221, miR-222, miR-146a, and miR-146b expression has been also associated with high-risk PTC features [18,42,43]. Increased expression of miR-221 and miR-222 is implicated in increased tumor size and increased probability of angioinvasion, lymph node and distant organ metastasis [42,44]. OncomiRs miR-221 and miR-222 were also associated with less differentiated tumors and reported to play a role in PTC aggressiveness [37]. Moreover, the downregulation of the expression of miR-21 and miR-9 has been implicated in the recurrence of PTC tumors [19]. Downregulation of miR-9 has an effect on cyclin D1 overexpression and p27 underexpression, which in turn predicts lymph node metastasis in PTC [45]. Downregulation of miR-21 affects overexpression of intercellular adhesion molecule-1 (ICAM1). This results in increased PTC aggression manifested, for example, by lymph node metastasis [46].

Furthermore, the miRNA expression pattern bears potential as a PTC biomarker. Park et al. demonstrated three miRNAs (miR-136, miR-21, miR-127) as diagnostic and prognostic markers of PTC [47]. Zei et al. in their study demonstrated that miR-155 causes downregulation of sex-determining region Y-box 17 (Sox17) and overexpression of inflammatory cytokine interleukin (IL22). This promotes the migration of PTC cells [48]. Another biomarker that could differentiate PTC from benign lesions is miR-155 [36]. Moreover, the let-7 family, which downregulates *RAS* expression, has the potential to be a biomarker of PTC [49]. Geraldo et al. point out that let-7f, in combination with miR-146-5p, can be a prognostic tool in PTC [50]. In another study, Ma et al. showed that miR-199a-5p acts by inhibiting snail family zinc finger 1 (*SNAI1*), resulting in SNAl1 overexpression and increased PTC proliferation [51]. In a recent study, downregulation of miR-363-3p was found to inhibit PTC progression by targeting NIN1/RPN12 binding protein 1 (*NOB1*) [52]. In contrast, Wang et al. showed that downregulation of miR-599 promotes PTC cell proliferation and inhibits apoptosis by targeting *HEY2* gene expression [53]. PTC’s pathogenesis can be further determined using miRNA evaluation. This may provide insights for determining potential medical targets [54]. Moreover, specific miRNAs may be considered PTC biomarkers [55]. Introducing an miRNA panel to routine diagnostics may improve the accuracy of the obtained FNAB results [56]. Moreover, the determination of the miRNA profile would allow personalization of the treatment strategy as well as the determination of the individual risk of cancer progression or metastasis [57]. As such, our purpose in this study was to identify DEMs in PTC with potential diagnostic utility.

In our study, we identified ten miRNAs (miR-146b-5p, miR-221-3p, miR-221-5p, miR-34-5p, miR-551b-3p, miR-152-3p, miR-15a-5p, miR-31-5p, and miR-7-5p) that were differentially expressed in PTC compared to normal thyroid tissue. All the DEMs included in the study are implicated in PTC development or cancer progression. In this case, the determination of DEMs during PTC can provide insights into potential target genes and metabolic pathways with clinical usefulness. It has been demonstrated that the overexpression of miR-146b-5p in PTC promotes invasion and metastasis and induces epithelial–mesenchymal transition by targeting zinc RING finger 3 *(ZNRF3*) [58,59]. Furthermore, the deregulation of miR-221-3p expression is involved in the regulation of the suppressor of cytokine signaling 3/signal transducer and activator of transcription 3 (*SOCS3/STAT3*) pathway, which is particularly relevant in cancer resistance. In this case, radiosensitivity in thyroid cancer can be reduced by targeting the influence of miR-221-3p on the solute carrier family 5 member 5 (*NIS*). [60]. Furthermore, miR-34a can exert an inhibitory effect related to cell proliferation by targeting many protooncogenes, implicating it in tumorigenesis and cancer progression [61]. Long et al. showed that miR-34a expression is associated with PTC’s tumor stages, histopathological types, and fluorodeoxyglucose maximum standardized uptake value [62]. Accordingly, miR-551b-3p was also shown to relate to PTC’s clinicopathological features [14]. Moreover, the downregulation of miR-152-3p, considered a tumor suppressor, promotes PTC development [63]. Kang et al. found that miR-152-3p overexpression was implicated in *ERBB3* downregulation, which inhibits human PTC (TPC-1) cell proliferation. The overexpression of miR-221 inhibited reversion-inducing cysteine-rich protein with Kazal motif (*RECK*), which is involved in promoting the invasiveness and migration of PTC cells [44]. Wang et al. found that miR-15a-5p promoted the growth of PTC cells by regulating hexokinase 2 (*HK2)* expression, and suggested it as a potential PTC therapeutic target [64]. Rosignolo et al. showed the association of miR-31-5p overexpression with a higher risk of PTC tumor recurrence [34]. Augenlicht et al. indicated that the tumor suppressor miR-7-5p inhibits thyroid cancer cell proliferation and that its target genes inhibit the EGFR/MAPK and IRS2/PI3K signaling pathways. The effect of miR-7-5p downregulation may promote PTC proliferation and invasiveness [65], and specific miRNA expression patterns may be clinically useful for tailoring treatment strategies as a central feature of miRNA-based treatments for cancer and cancer management in the future [66].

Additionally, we identified key DEM target genes, established the functional enrichment genes, and constructed a PPI network; we specifically identified *MAP2K1*, *RAF1*, *IRS1*, *EGFR*, *KIT*, *KDR*, *ERBB3*, *NRAS*, *PIK3R1*, and *PIK3CB* as hub genes. The product of the *MAP2K1* gene is a serine/threonine and tyrosine kinase, which is, in turn, activated by phosphorylation through the action of RAF kinase [67]. *RAF1* participates in the RAS/MEK/ERK signaling pathway. Moreover, *RAF1* was observed to be upregulated in PTC development [68]. Specifically, *RAF*1 is an important factor promoting tumorigenesis and PTC tissue progression. In a similar study by Li et al., patients with PTC showed elevated *RAF-1* and decreased miR-485-5p expression [69]. Chen et al. showed that miR-1271 inhibited PTC development by affecting *IRS1*, implicated in epithelial–mesenchymal transition and the phosphatidylinositol 3-kinase (PI3K)/protein kinase B (AKT) pathway [70]. Additionally, Yang et al. proved the importance of the PI3/AKT pathway in their study, which showed a frequent association with tumor progression and resistance to cancer therapies as the pathway’s activity increased [71]. In turn, PIK3R1 and PIK3CB are the components of the previously mentioned PIK3/AKT/mTOR signaling pathway [72]. Moreover, the GAS1 gene and PI3/AKT pathway are connected. Growth arrest specific 1 (*GAS1*), a target gene of miR-34a, leads to the inhibition of apoptosis in PTC and an increase in PTC cell proliferation. [73]. Modeling miRNA expression as a medical target may lead to the complete inhibition of tumor progression. Furthermore, EGFR is a tyrosine kinase implicated in cell proliferation. Masago et al. described the effect of *EGFR*-activating mutations on PTC development [74]. Inhibitors of ERRB3, which is also known as a proto-oncogene and a member of the EGFR family, may be effective in the treatment of PTC [75]. Alternatively, the oncogene *KIT*, described for its role in PTC and other cancers, encodes a receptor tyrosine kinase that affects cellular growth and differentiation [76]. The dysregulation of miR-221, miR-222, and miR-146 has been implicated in *KIT* downregulation [77]. *KDR* is a tyrosine kinase whose overexpression in PTC was described [78]. *NRAS* encodes the N-ras protein responsible for regulating cell division. The importance of *NRAS* mutations in PTC was also demonstrated [79]. Novel transcriptomic approaches specifically targeting PTC hub genes should be further investigated. Moreover, overcoming drug resistance in cancer therapy represents an increasing clinical challenge; thus, the results obtained from the present research may lead to the discovery of the molecular mechanism underlying cancer development and progression.

In this study, through KEGG analysis, we identified the most relevant pathways in the pathogenesis of PTC, i.e., EGFR tyrosine kinase inhibitor resistance, the mTOR signaling pathway, and the ERBB3 signaling pathway, all important in modulating the response to anticancer treatment [74,75,80]. Tavares et al. implicated the mTOR pathway in distant metastasizing and therapy resistance in PTC [81]. The recent literature suggests that the potential inhibition of mTOR activity may prevent cancer progression and improve the survival prognosis for patients after treatment [82]. Moreover, targeting ERBB3 and EGFR signaling [83,84,85] is an effective method for overcoming cancer progression and resistance to anticancer therapy [85,86].

Using the GO approach, biological processes were significantly enriched for nervous system development and several metabolism-related terms. Neurogenesis, similar to angiogenesis, is also an important modulator of cancer cells progression [87]. Literature data demonstrate the presence of nerves in the tumor microenvironment and their promotion of tumorigenesis and disease progression [88]. GO molecular function analysis revealed that among the discovered DEGs were those involved in kinase activity and protein binding. Our study demonstrates key molecular pathways and provides much needed insights into potential targets and treatment options of PTC.

All the identified DEMs showed high diagnostic utility (excluding miR-31-5p). Possible clinical usefulness in PTC diagnosis was observed for miR-146-5p (AUC = 0.770), miR-551-3p (AUC = 0.740), and miR-222-3p (AUC = 0.720). Specifically, miR-146-5p (AUC = 0.770) expression had the highest diagnostic utility. To increase the diagnostic accuracy of using miRNAs, we assessed a combination of four miRNAs (miR-152-3p, miR-221-3p, miR-551b-3p, and miR-7-5p) with a calculated AUC of 0.841. Moreover, the AUC values obtained in our study are comparable to those previously published in the literature [16,36,47]. Other authors also proposed miR-221 measurement as a potential biomarker of PTC recurrence [18]. Duan et al. showed that miR-7-5p and miR-451 can be considered diagnostic biomarkers of PTC [89]. Additionally, the individual miRNAs that comprise our proposed panel have been previously mentioned as potential biomarkers of PTC [14,89,90,91]. The overexpression of these miRNAs reflects recent results obtained by Qiao et al., who demonstrated that the combination of five miRNAs, including miR-1296-5p, miR-1301-3p, miR-532-5p, miR-551b-3p, and miR-455-3p, is potentially useful in the diagnosis of PTC (AUC of 0.941 with 82.14% sensitivity and 100% specificity in their study) [14]. The lack of a sensitive panel may limit the introduction of a miRNA diagnostic panel into clinical routines. Moreover, cost–benefit and cost-effectiveness analyses are required. Nevertheless, the procedure’s cost will consistently decrease with the development of molecular techniques. Introducing a diagnostic miRNA panel in PTC may result in more accurate diagnoses in difficult and questionable cases, enable more accurate therapeutic decisions, help to prioritize patients who require special clinical management or more aggressive treatment, and be part of a more personalized approach to medicine. Lastly, and in reference to the heterogeneity of PTCs, the introduction of a diagnostic panel rather than a separate measurement would be preferable. In order for the miRNA panel to be applied in clinical practice, though, it is important to evaluate the miRNA presence in FNAB and biofluids.

The results of our study are in line with the current knowledge about miRNAs in the pathogenesis of PTC. The theranostic utility of miRNAs and their prognostic potential in PTC have been demonstrated [16]. However, our analyses are of crucial importance for the exploration of the role of miRNA expression patterns in the pathogenesis of PTC. Furthermore, general studies of PTC have aimed to establish novel noninvasive diagnostic panels, but the results have been insufficient. Therefore, we proposed a miRNA PTC diagnostic panel consisting of miR-152-3p, miR-221-3p, miR-551b-3p, and miR-7-5p. The wide range of studied miRNAs and the validation of the results obtained by PCR assays constitute the strengths of this study. However, because of the limited size of the experimental group, further evaluation and data validation using a larger cohort are required to confirm the diagnostic usefulness of the proposed miRNA panel.

## 5. Conclusions

Our analysis demonstrates the usefulness of evaluating miRNA expression patterns in PTC diagnosis. Moreover, a four-miRNA combination—miR-152-3p, miR-221-3p, miR-551b-3p, and miR-7-5p—may be introduced as a diagnostic panel in PTC patients. In addition, the characterization of the regulatory network of the DEMs and target genes will be important for investigating the pathways driving disease progression. Nevertheless, additional studies are necessary for understanding the role of dysregulated miRNAs in the molecular mechanisms underlying PTC pathogenesis.

## Figures and Tables

**Figure 1 cancers-14-02679-f001:**
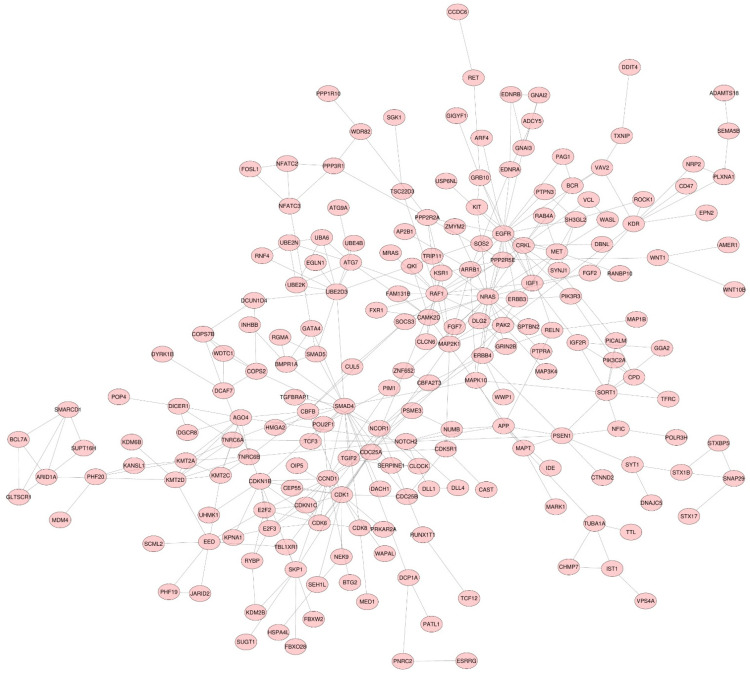
Target genes of DEMs rendered using STRING database. Only connected nodes are present. Only interactions with high confidence interaction scores (>0.9) are shown. The PPI enrichment value was predicted to be 8.44 × 10^−15^.

**Figure 2 cancers-14-02679-f002:**
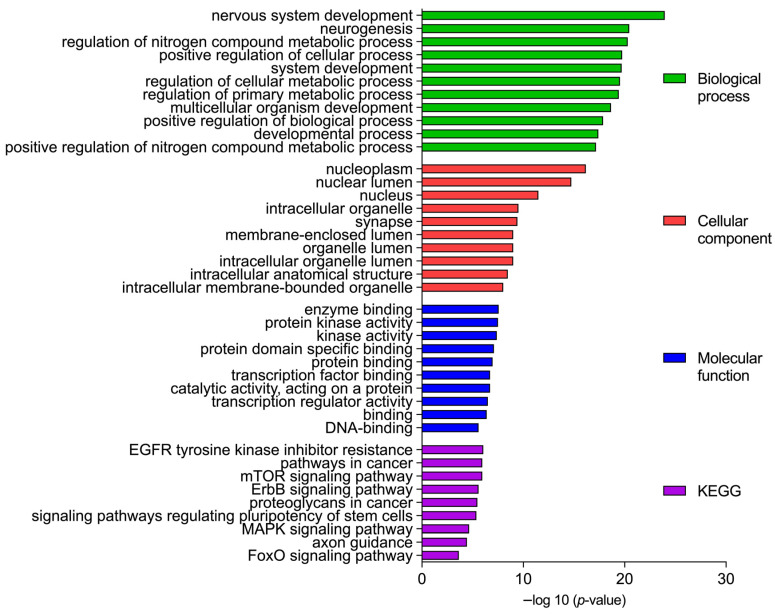
Gene Ontology (GO) enrichment analysis. Top 10 significantly enriched GO (−log10 (*p*-value)) categories of the target genes in the cellular components, molecular function, biological processes, and KEGG enrichment. KEGG, Kyoto Encyclopedia of Genes and Genomes.

**Figure 3 cancers-14-02679-f003:**
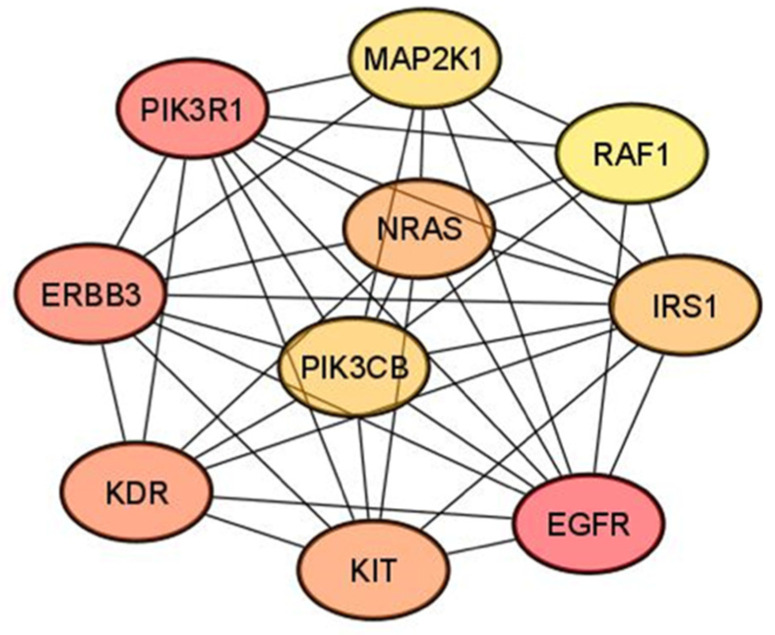
The networks of the top 10 hub genes.

**Figure 4 cancers-14-02679-f004:**
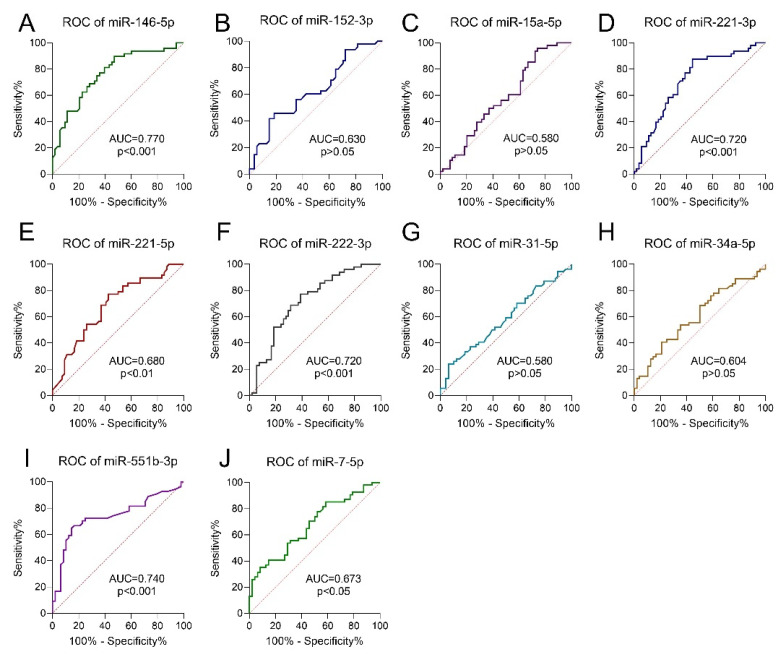
ROC analysis was conducted to evaluate the diagnostic value of the DEMs as diagnostic biomarkers of PTC vs. control: (**A**) miR-146-5p; (**B**) miR-152-3p; (**C**) miR-15a-5p; (**D**) miR-221-3p; (**E**) miR-221-5p; (**F**) miR-222-3p; (**G**) miR-31-5p; (**H**) miR-34a-5p; (**I**) miR-551b-3p; (**J**) miR-7-5p. AUC values were calculated to estimate diagnostic performance of the DEMs in PTC. *p*-values indicate a significant difference from AUC = 0.5 (borderline of the diagnostic usefulness of the test).

**Figure 5 cancers-14-02679-f005:**
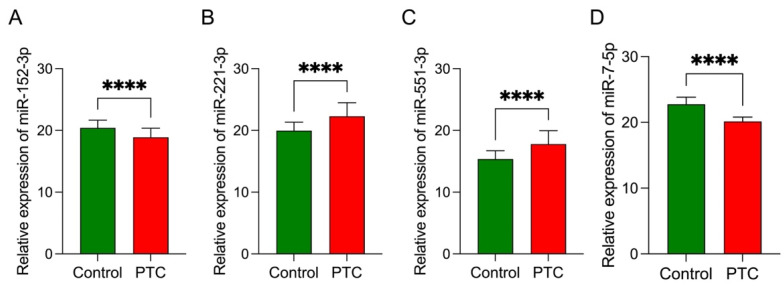
Relative expression of DEMs included in the logistic regression model in control (N = 39) and PTC samples (N = 41): (**A**) miR-152-3p (Fold change (FC) = 0.37); (**B**) miR-221-3p (FC = 5.78); (**C**) miR-551b-3p (FC = 5.68); (**D**) miR-7-5p (FC = 0.18). Each bar represents the geometric mean ± standard error of mean of the ratio of miRNA expression and reference miRNAs (miR-103a-3p and U6 snRNA) calculated using the qBase MSExcel VBA based on multiple samples and multiple reference miRNAs. Asterisks indicate the levels of significance of differences (**** *p*< 0.0001); Mann–Whitney U-test was used to compare PTC and control samples. PTC, papillary thyroid cancer. FC values were provided using the GeneGlobe Data Analysis Center.

**Table 1 cancers-14-02679-t001:** Characteristics of the study group.

Characteristic	Samples of PTC
Total	41
Men/women	8/33
pT1	31
pT2	5
pT3	2
pT4	3
Men: Age at diagnosis—mean (range)	59.7 (41–77)
Women: Age at diagnosis—mean (range)	53.1 (30–77)
Diameter of the tumor—mean (mm)	10.5
Number of samples with features of angioinvasion	12
Number of samples with multifocal features	27

PTC- Papillary Thyroid Cancer.

**Table 2 cancers-14-02679-t002:** miRNAs with significantly different expression between PTC and the control group (FDR < 0.05 and |FC| ≥ 1.5).

miRNA	FC	*p*-Value	FDR
miR-146b-5p	6.62	0.00000001	0.00
miR-221-3p	3.23	0.00000001	0.00
miR-221-5p	2.36	0.00000016	0.00
miR-222-3p	2.94	0.00000001	0.00
miR-34a-5p	1.65	0.00002568	0.00
miR-551b-3p	2.63	0.00000003	0.00
miR-152-3p	−1.53	0.00040006	0.01
miR-15a-5p	1.57	0.00031352	0.01
miR-31-5p	1.60	0.00051338	0.02
miR-7-5p	−2.57	0.00108612	0.03

FC, fold change; FDR, false discovery rate.

**Table 3 cancers-14-02679-t003:** Top 10 hub genes.

Hub Gene	DEMs Targeting Hub Gene
*MAP2K1*	miR-152-3p; miR-15a-5p; miR-34a-5p
*RAF1*	miR-15a-5p; miR-7-5p
*IRS1*	miR-15a-5p; miR-7-5p
*EGFR*	miR-152-3p; miR-221-3p; miR-222-3p; miR-7-5p
*KIT*	miR-152-3p; miR-221-3p; miR-222-3p
*PIK3CB*	miR-146b-5p; miR-7-5p
*NRAS*	miR-146b-5p; miR-152-3p
*KDR*	miR-15a-5p; miR-221-3p; miR-222-3p
*ERBB3*	miR-152-3p; miR-221-3p; miR-222-3p
*PIK3R1*	miR-15a-5p; miR-221-3p; miR-222-3p

*MAP2K1*, mitogen-activated protein kinase 1; *RAF1*, Raf-1 proto-oncogene, serine/threonine kinase; *IRS1*, insulin receptor substrate 1; *EGFR*, epidermal growth factor receptor; *KIT*, KIT proto-oncogene receptor tyrosine kinase; *KDR*, kinase insert domain receptor; *ERBB3*, Erb-B2 receptor tyrosine kinase 3; *NRAS*, NRAS proto-oncogene, GTPase; *PIK3R1*, phosphoinositide-3-kinase regulatory subunit 1; *PIK3CB*, phosphatidylinositol-4,5-bisphosphate 3-kinase catalytic subunit beta.

**Table 4 cancers-14-02679-t004:** Summary of the basic parameters and common quality measures of the models.

Model	TP Rate	FP Rate	Precision	AUC	Intercept	Coefficient
x_1_ = miR-152-3p	0.769	0.227	0.774	0.841	−1.8033	a_1_ = −1.2104
x_2_ = miR-221-3p						a_2_ = 0.8173
x_3_ = miR-551b-3p						a_3_ = 0.5172
x_4_ = miR-7-5p						a_4_ = −0.0178

TP, true positive; FP, false positive; AUC, area under the receiver operating characteristic curve.

## Data Availability

The NanoString data were deposited in the Gene Expression Omnibus (GEO) database (GSE191117) link: https://www.ncbi.nlm.nih.gov/geo/query/acc.cgi?acc=GSE191117.

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
