# Peer review of "Expression Profile and Diagnostic Significance of MicroRNAs in Papillary Thyroid Cancer"

_cancers, 2022, doi:10.3390/cancers14112679_

Round 1

Author Response

Answer to Reviewer 1

Dear Reviewer, thank you for your constructive comments concerning our manuscript entitled ‘Expression Profile and Diagnostic Significance of MicroRNAs in Papillary Thyroid Cancer’. We have studied your comments carefully and made major correction which we hope meet with your approval.
Our answers to your points are as follows.
1. In my opinion, the most fundamental issue with this work is that it is written in such a way as no one had done this kind of study before.
Response: The manuscript has been rewritten according to the suggestions.
2. Technical questions regarding the Nanostring.
Response: High expression of miR-146b-5p observed in our analysis agrees with available literature data provided by the Reviewer. The difference between AUC equal 0.77 and 0.90 (predicted by the Reviewer) for a specific miRNA expression could be interpreted in various ways. Nevertheless, the method (small RNA sequencing, microarray, PCR), type of specimen used in the analyzes (FFPE, FNAB, frozen tissue), inclusion/exclusion criteria (comorbidities, metastases), type of reference tissue (paired from cancer patients, normal tissue from healthy patients, benign tissue), and histological type of cancer should also be considered when comparing and interpreting the results. What should be noted is that most articles cited by the Reviewer are based on PCR method. Moreover, our preliminary study showed that metastasis significantly affect miRNA profile, thus, we decided to exclude metastatic patients from the study.
3. Raw data interpretation.
Response: The most fundamental aspect of Nanostring analysis is that this method does not measure the number of copies as stated by the Reviewer. This method uses molecular barcodes and single molecule imaging to detect and count hundreds of miRNAs or genes using specific panels. What is most important, both NanoString (which counts single molecules with no amplification step) and PCR (which utilizes an amplification method) provided similar expression patterns in our study.
When assessing our results, no flag was present in any step of QC based on the four default parameters (binding density, imaging, positive control, ligation) in nSolver software provided by the Nanostring manufacturer proving the correctness of ligation, hybridization, and scanning processes. Thus, we do not see any basis to question our raw data. Raw data alone (checking the counts for specific miRNAs without any normalization step) do not provide any information on miRNA expression. The nCounter program uses a normalization factor to account for technical noise (such as variations in hybridization, purification, binding efficiency). The normalization factor was generated using the geometric mean of the top 100 miRNAs for each sample; raw counts were multiplied by this sample specific normalization factor to produce the normalized data. In addition, six internal negative controls and six positive controls were included in the nCounter miRNA Expression assays, to correct for background noise. The mean of the negative controls served as the medium stringency threshold for miRNA detection and high stringency was calculated by adding 2SD to the mean of the negative controls (threshold = mean + 2SD).
In summary, we would like to thank you once again for such an accurate analysis of our manuscript and all the valuable recommendations.
Sincerely,

Mariusz Rogucki

Department of Endocrinology, Diabetology and Internal Medicine Medical University of Bialystok, 15-276 Bialystok, Poland

Reviewer 2 Report

In this manuscript, Rogucki and colleagues have reported the investigation of miRNA expression profile and network in papillary thyroid cancer (PTC) tissues. This work has any diagnostic benefit of several miRNA candidates and their networks which were expected to be upregulated and active in the PTC tissues. Some data may contain a value addressing an insight into PCT diagnostic biomarker discovery. Here are my comments and concerns below.

  • Regarding the results of Table 2, what does the control group define? Is it RNA from 39 reference tissues with no malignancy (lines 94-95)? Clarify more about the controls used. Why is the p-value all “0.00”? The authors have stated that serum 798 miRNAs were profiled for tumor-specific miRNA expression (lines 159-160). Are the data from the circulating miRNAs of PTC patients available to show?
  • Exactly which miRNAs were used as DEMs to predict their target genes in lines 168~ and Figure 1? Are they the miRNAs shown in Table 2? The authors should specify them. In addition, Figure 1 doesn’t sound clear on circle color difference and should include legends with ample explanation.
  • In lines 175-187, authors have described functional enrichment analysis using GO and KEGG enrichment tools. The authors have described about any implication of KEGG to PTC. Briefly discuss about what implications to the higher p values’ catagories (biological process, cellular component, molecular function) associate with PTC, as well. Also, an implicative remark with a sentence is recommended to include in the end of the paragraph (line 187~).
  • Related to top 10 hub genes networks (regarding Figure 3), authors have discussed on their possible scientific implication to PTC. It would be better to specifically define the DEMs that target those 10 hub genes in the text.
  • Please include more information on the data of Figure 4 in its legends.
  • About logistic regression model (line 215~), four DEMs show pretty high value upon combined. Authors may wish to explain more about why such results are generated, which may be helpful for general readership.
  • QPCR results (Figure 5) of 4 DEMs selected show significant alterations between control and PTC tissues. Are the data (bar graphs) indicating expressions relative to controls (miR-103-3p and U6 snRNA)? Why were two endogenous genes used? Were both or either one used? Are they the values of 2^-dCT? The “N” (sample number) per result (e.g., bar) needs to be stated. The range of values for the relative expression is 15~25, which is quiet unusual. Authors should include more descriptions about PCR methodology and results if available.
  • Clarify more the definitions of AUC in the Abstract. To my knowledge, the receiver operating characteristic curve should be abbreviated to ROC curve, instead of AUC (lines 38, 135, etc.). The authors need to correct it. needs to come to understanding and construe grammatically. It should be rewritten.
  • What is “EC” diagnosis in line 39? Please spell it out when it comes first in the text.

Author Response

Answer to Reviewer 2
Dear Reviewer, thank you for your constructive comments concerning our manuscript entitled ‘Expression Profile and Diagnostic Significance of MicroRNAs in Papillary Thyroid Cancer’. We have studied your comments carefully and made major correction which we hope meet with your approval.
Our answers to your points are as follows.
1. Regarding the results of Table 2, what does the control group define? Is it RNA from 39 reference tissues with no malignancy (lines 94-95)?
Response: Control group consisted of normal tissue (confirmed by a pathologist) obtained from PTC patients. This information has been included in the manuscript (section Study Subjects: ‘The control group consisted of specimens from the same patients with normal nontumor tissues.’).
2. Why is the p-value all “0.00”?
Response: This value ‘0.00’ means that the FDR value is less than 0.01.
3. The authors have stated that serum 798 miRNAs were profiled for tumor-specific miRNA expression (lines 159-160). Are the data from the circulating miRNAs of PTC patients available to show?
Response: We assessed the miRNA profile expression in FFPE tissues – in PTC and reference normal thyroid tissue. The word ‘serum’ appeared by mistake. This has been corrected and the word ‘serum’ has been replaced by ‘tissue’.
4. Exactly which miRNAs were used as DEMs to predict their target genes in lines 168~ and Figure 1? Are they the miRNAs shown in Table 2? The authors should specify them.
Response: In our study we found ten DEMs (which are shown in Table 2) and they were used to predict target genes. This information has been specified in the manuscript.
5. In addition, Figure 1 doesn’t sound clear on circle color difference and should include legends with ample explanation.
Response: The quality of this figure has been improved.
6. In lines 175-187, authors have described functional enrichment analysis using GO and KEGG enrichment tools. The authors have described about any implication of KEGG to PTC. Briefly discuss about what implications to the higher p values’ categories (biological process, cellular component, molecular function) associate with PTC, as well. Also, an implicative remark with a sentence is recommended to include in the end of the paragraph.
Response: The improvements have been made according to the suggestion.
7. Related to top 10 hub genes networks (regarding Figure 3), authors have discussed on their possible scientific implication to PTC. It would be better to specifically define the DEMs that target those 10 hub genes in the text.
Response: The information about the DEMs targeting hub genes has been included in the manuscript (Table 3).
8. Please include more information on the data of Figure 4 in its legends.
Response: This has been improved according to the suggestion.
9. About logistic regression model (line 215~), four DEMs show pretty high value upon combined. Authors may wish to explain more about why such results are generated, which may be helpful for general readership.
Response: Logistic regression is a supervised learning algorithm commonly used for binary classification tasks. Intercept and coefficients are parts of the model and are provided to use this model in future. Intercept is the log odds when the variable is 0, this means the patients with no PTC. Coefficients describe how much the log odds change with the presence of PTC. True positive, false positive, precision, and AUC values describe the
diagnostic usefulness of the model. This has been included in the manuscript according to the Reviewer’s suggestion.
10. QPCR results (Figure 5) of 4 DEMs selected show significant alterations between control and PTC tissues. Are the data (bar graphs) indicating expressions relative to controls (miR-103-3p and U6 snRNA)? Why were two endogenous genes used? Were both or either one used? Are they the values of 2^-dCT? The “N” (sample number) per result (e.g., bar) needs to be stated. The range of values for the relative expression is 15~25, which is quiet unusual. Authors should include more descriptions about PCR methodology and results if available.
Response: the qBase analyzer converts the Ct values from all runs within one experiment to normalized and rescaled quantities that can be visualized in graphs. The x asis of our PCR figure presents the relative normalized expression of studied miRNAs compared to reference miRNAs (two housekeeping miRNAs stably expressed in the PTC cells). The article (Hellemans, J., Mortier, G., De Paepe, A. et al. qBase relative quantification framework and software for management and automated analysis of real-time quantitative PCR data. Genome Biol 8, R19 (2007). https://doi.org/10.1186/gb-2007-8-2-r19) by Hellemans, a CEO and founder of qBase is cited 2751 times (checked 25.04.2022) which proves the usefulness and robustness of the qBase calculation.
Additional information has been included in the methodology section. Sample number has been provided in the figure legend.
11. Clarify more the definitions of AUC in the Abstract. To my knowledge, the receiver operating characteristic curve should be abbreviated to ROC curve, instead of AUC (lines 38, 135, etc.). The authors need to correct it. needs to come to understanding and construe grammatically. It should be rewritten.
Response: In our opinion the sentence ‘The highest area under the receiver operating characteristic curve (AUC) for DEMs…’ is correct. It means the area under the ROC (in full name) curve. However, we agree that the sentence should be rewritten. Thus, it has been corrected according to the suggestion.
12. What is “EC” diagnosis in line 39? Please spell it out when it comes first in the text.
Response: ‘EC’ is an author’s mistake. This has been corrected to ‘PTC’ in the abstract.
In summary, we would like to thank you once again for such an accurate analysis of our manuscript and all the valuable recommendations.
Sincerely,

Mariusz Rogucki Department of Endocrinology, Diabetology and Internal Medicine Medical University of Bialystok, 15-276 Bialystok, Poland

Reviewer 3 Report

The authors have evaluated how investigating miRNA expression pattern may aid the diagnosis of papillary thyroid cancer. Their purpose was primarily to supplement and increase the diagnostic accuracy of fine needle aspiration biopsy in this regard.

The authors have used FFPE materials from surgical specimens of PTC and used non-tumor tissues as controls. A pathologist has identified tumor-rich regions of interest in the FFPE blocks. Interestingly, metastatic PTC cases were excluded. Exclusion criteria also included other comorbidities as well as cigarette smoking.

The authors have identified differently expressed miRNAs when PTCs were compared to controls by Nanostring. The authors have validated their findings with RT-PCR. The differently expressed miRNAs were mostly recognized as relevant in PTC earlier in the literature. The miRNAs are involved in pathways important in the pathogenesis of this disease.

The authors have designed a panel of miRNAs that may be used as a diagnostic tool with high accuracy.

A conceptual flaw of the study renders the results somewhat misleading. The purpose of the authors was to design a diagnostic tool to supplement FNAB for recognizing PTC. FNAB of PTC would have an extremely high accuracy when normal thyroid gland is compared to PTC, however, that is not the usual diagnostic situation. PTC is compared to other thyroid gland lesions.

The authors have used non-neoplastic tissue (presumably of the thyroid gland) as controls. They should have used benign thyroid gland lesions as controls. Also, the authors have used FFPE material in their study, however, FNAB uses non-FFPE material. Performance of the miRNA profiling may prove to be different.

Due to the above, the manuscript should be revised. The authors should explain that they have conducted a pilot study to see if normal thyroid gland can be differentiated from PTC using their Nanostring-based approach and discuss those findings appropriately. It should not be stated that the purpose was to establish a diagnostic tool, because that goal has not been reached. The appropriate interpretation of accuracy and precision they report should be discussed to avoid confusion.

Further adjustments should be considered as well:

The authors should not state that a noninvasive miRNA PTC diagnostic panel was designed (line 365). Their panel needs thyroid gland sampling, which can only be performed using invasive techniques.

The abstract does not explain what EC stands for (line 39).

The authors should explain whether the controls were representing normal thyroid gland tissue or some other pathology. Tumor-content of the slides used in the study should be explained. If possible, morphological data of the cases (morphological type, pT stage, etc.) should be included.

Results, line 159-160: The authors should explain why serum expression of miRNAs was measured and the serum of whom was measured. It is not clarified in the Methods section.

What was the unit of relative expression on Figure 5? Are these delta-Ct values? Fold change acquired by RT-PCR should be stated at least in the figure legend.

Why were metastatic PTC cases excluded? If the authors mean that only PTC samples from the thyroid gland were analyzed, they should clarify, if any case that should metastasis clinically were excluded, the reason should be explained.

The authors should discuss why other comorbidities or smoking was excluded and how this may affect diagnostic performance of miRNA profiling.

Author Response

Answer to Reviewer 3
Dear Reviewer, thank you for your constructive comments concerning our manuscript entitled ‘Expression Profile and Diagnostic Significance of MicroRNAs in Papillary Thyroid Cancer’. We have studied your comments carefully and made major correction which we hope meet with your approval.
Our answers to your points are as follows.
1. A conceptual flaw of the study renders the results somewhat misleading. The purpose of the authors was to design a diagnostic tool to supplement FNAB for recognizing PTC. FNAB of PTC would have an extremely high accuracy when normal thyroid gland is compared to PTC, however, that is not the usual diagnostic situation. PTC is compared to other thyroid gland lesions.
Response: We agree with this comment. However, our preliminary study involved only PTC tissues and non-neoplastic reference tissue obtained from the same patients. In our study, we aimed to identify miRNAs, or miRNA correlations, that are most specific for PTC. Therefore, we included tissues containing at least 50% of the surface area of FFPE preparations in the study group and tissues not containing PTC cells in the control group. All tissues were confirmed by a pathomorphologist. Previously, Chen et al. (MicroRNA analysis as a potential diagnostic tool for papillary thyroid carcinoma. DOI: 10.1038/modpathol.2008.105) analyzed research material from FNAB and FFPE tissues, finding higher quality miRNAs in FFPE tissues. Our ongoing study also aims to include benign lesions within the control group.
2. The authors have used non-neoplastic tissue (presumably of the thyroid gland) as controls. They should have used benign thyroid gland lesions as controls. Also, the authors have used FFPE material in their study, however, FNAB uses non-FFPE material. Performance of the miRNA profiling may prove to be different.
Response: The answer, to this fair point, is contained above.
Due to the above, the manuscript should be revised. The authors should explain that they have conducted a pilot study to see if normal thyroid gland can be differentiated from PTC using their Nanostring-based approach and discuss those findings appropriately. It should not be stated that the purpose was to establish a diagnostic tool, because that goal has not been reached. The appropriate interpretation of accuracy and precision they report should be discussed to avoid confusion.
Response: As mentioned above, the study was designed using tissues without PTC cells and without benign lesions as a control group to initially develop a diagnostic panel. We recognize that the issue requires further research, which we are currently performing.
3. The authors should not state that a noninvasive miRNA PTC diagnostic panel was designed (line 365). Their panel needs thyroid gland sampling, which can only be performed using invasive techniques.
Response: This sentence has been corrected.
4. The abstract does not explain what EC stands for (line 39).
Response: ‘EC’ is an author’s mistake. This has been corrected to PTC in the text.
5. The authors should explain whether the controls were representing normal thyroid gland tissue or some other pathology. Tumor-content of the slides used in the study should be explained. If possible, morphological data of the cases (morphological type, pT stage, etc.) should be included.
Response: The tissues, constituting the study group, were checked by a pathomorphologist and contained at least 50% of the surface area of the whole preparation. This information has been included in the Materials section.
6. Results, line 159-160: The authors should explain why serum expression of miRNAs was measured and the serum of whom was measured. It is not clarified in the Methods section.
Response: We assessed the miRNA profile expression in FFPE tissues – in PTC and reference normal thyroid tissue. Word ‘serum’ appeared by mistake and has been replaced by ‘tissue’.
7. What was the unit of relative expression on Figure 5? Are these delta-Ct values? Fold change acquired by RT-PCR should be stated at least in the figure legend.
Response: the qBase analyzer converts the Ct values from all runs within one experiment to normalized and rescaled quantities that can be visualized in graphs. The x asis of our PCR figure presents the relative normalized expression of studied miRNAs to reference miRNAs (two housekeeping miRNAs stably expressed in the PTC cells). The article (Hellemans, J., Mortier, G., De Paepe, A. et al. qBase relative quantification framework and software for management and automated analysis of real-time quantitative PCR data. Genome Biol 8, R19 (2007). https://doi.org/10.1186/gb-2007-8-2-r19) by Hellemans, a CEO and founder of qBase is cited 2751 times (checked 25.04.2022) which proves the usefulness and robustness of the qBase calculation.
Fold change values for RT-PCR result have been provided in the legend of Figure 5 (miR-152-3p, FC=0.37; miR-221-3p, FC=5.78; miR-551-3p, FC=5.68; miR-7-5p, FC=0.18) using the GeneGlobe Data Analysis Center (QIAGEN; geneglobe.qiagen.com).
8. Why were metastatic PTC cases excluded? If the authors mean that only PTC samples from the thyroid gland were analyzed, they should clarify, if any case that should metastasis clinically were excluded, the reason should be explained.
Response: We performed additional analyses in PTC tissues and compared PTC tissues from patients with and without metastasis. We found significant differences in miRNA expression pattern. Thus, considering additional comorbidities influence miRNA profile, we decided to exclude metastatic patients from the study.
9. The authors should discuss why other comorbidities or smoking was excluded and how this may affect diagnostic performance of miRNA profiling.
Response: As we cannot predict the influence of other comorbidities on miRNA expression profile, we decided to exclude the patients with any additional disorders. Moreover, according to the research by Willinger et al., (https://doi.org/10.1161/CIRCGENETICS.116.001678) smoking significantly affects miRNA profile. MiRNA signature of cigarette smoking and altered expression of inflammatory mediators was demonstrated recently. Cigarette smoke may also cause dysregulation by affecting regulatory mechanisms controlling miRNA expression. Thus, we also excluded smoking patients from the study.
In summary, we would like to thank you once again for such an accurate analysis of our manuscript and all the valuable recommendations.
Sincerely,

Mariusz Rogucki Department of Endocrinology, Diabetology and Internal Medicine Medical University of Bialystok, 15-276 Bialystok, Poland

Round 2

Reviewer 1 Report

  1. In my opinion, the most fundamental issue with this work is that it is written in such a way as no one had done this kind of study before.

Response: The manuscript has been rewritten according to the suggestions.

Unfortunately, I cannot agree with this statement, I believe that adding two paragraphs is not enough. I can't help but think that the Introduction and Discussion need to be substantially rewritten to make it clear what has already been done in this area, what has been obtained in this study, and how these data relate to each other. I encourage the authors to make a new version of the manuscript, in which their research will be implemented in the existing knowledge system, and will not exist in parallel with it.

  1. Technical questions regarding the Nanostring.

Response: High expression of miR-146b-5p observed in our analysis agrees with available literature data provided by the Reviewer. The difference between AUC equal 0.77 and 0.90 (predicted by the Reviewer) for a specific miRNA expression could be interpreted in various ways. Nevertheless, the method (small RNA sequencing, microarray, PCR), type of specimen used in the analyzes (FFPE, FNAB, frozen tissue), inclusion/exclusion criteria (comorbidities, metastases), type of reference tissue (paired from cancer patients, normal tissue from healthy patients, benign tissue), and histological type of cancer should also be considered when comparing and interpreting the results. What should be noted is that most articles cited by the Reviewer are based on PCR method. Moreover, our preliminary study showed that metastasis significantly affect miRNA profile, thus, we decided to exclude metastatic patients from the study.

An interesting note about PCR, but I would like to see a more extensive discussion of this topic in the manuscript.

  1. Raw data interpretation.

Response: The most fundamental aspect of Nanostring analysis is that this method does not measure the number of copies as stated by the Reviewer. This method uses molecular barcodes and single molecule imaging to detect and count hundreds of miRNAs or genes using specific panels. What is most important, both NanoString (which counts single molecules with no amplification step) and PCR (which utilizes an amplification method) provided similar expression patterns in our study.

When assessing our results, no flag was present in any step of QC based on the four default parameters (binding density, imaging, positive control, ligation) in nSolver software provided by the Nanostring manufacturer proving the correctness of ligation, hybridization, and scanning processes. Thus, we do not see any basis to question our raw data. Raw data alone (checking the counts for specific miRNAs without any normalization step) do not provide any information on miRNA expression. The nCounter program uses a normalization factor to account for technical noise (such as variations in hybridization, purification, binding efficiency). The normalization factor was generated using the geometric mean of the top 100 miRNAs for each sample; raw counts were multiplied by this sample specific normalization factor to produce the normalized data. In addition, six internal negative controls and six positive controls were included in the nCounter miRNA Expression assays, to correct for background noise. The mean of the negative controls served as the medium stringency threshold for miRNA detection and high stringency was calculated by adding 2SD to the mean of the negative controls (threshold = mean + 2SD).

I do not understand the statement: “The most fundamental aspect of Nanostring analysis is that this method does not measure the number of copies…” The Nanostring technology involves counting barcodes that correspond to specific miRNAs (in this case) and thus counting copies of a specific miRNA in 100 ng of RNA isolated from a sample. Naturally, we are talking only about those miRNA copies to which the reporter probe was attached and was not washed out.

Also I disagree with the statement: “ Raw data alone (checking the counts for specific miRNAs without any normalization step) do not provide any information on miRNA expression.”

Raw data is quite informative about the expression of miRNAs, since you initially take approximately the same amount of RNA. For a more accurate comparison of samples, normalization is used, but the normalization factor is usually no more than 1.5. For this reason, I prefer raw data so as not to consider the fractional number of copies that result after normalization.

But my main question was that for some miRNAs that you consider in the manuscript, in 10-20% of samples (both cancer and normal) the values are at the level of negative controls, despite the fact that there are no QC flags. This situation would be worth analyzing, since it can significantly affect both the results and the conclusions.

Author Response

Dear Reviewer,

Thank you for your constructive comments concerning our manuscript entitled ‘Expression Profile and Diagnostic Significance of MicroRNAs in Papillary Thyroid Cancer’. We have studied your comments carefully and made major correction which we hope to meet your approval. Our answers to your points are as follows.

  1. Unfortunately, I cannot agree with this statement, I believe that adding two paragraphs is not enough. I can't help but think that the Introduction and Discussion need to be substantially rewritten to make it clear what has already been done in this area, what has been obtained in this study, and how these data relate to each other. I encourage the authors to make a new version of the manuscript, in which their research will be implemented in the existing knowledge system, and will not exist in parallel with it.
    Response: The Introduction and Discussion sections have been substantially improved. We decided to transfer the attribution to the current data to the discussion section, which is an extension of this topic. We hope now our study has been implemented in the current knowledge status.
    2. An interesting note about PCR, but I would like to see a more extensive discussion of this topic in the manuscript.
    Response: Extensive discussion of this topic have been introduced in the discussion section according to the suggestion.
    3. I do not understand the statement: “The most fundamental aspect of NanoString analysis is that this method does not measure the number of copies…” The NanoString technology involves counting barcodes that correspond to specific miRNAs (in this case) and thus counting copies of a specific miRNA in 100 ng of RNA isolated from a sample. Naturally, we are talking only about those miRNA copies to which the reporter probe was attached and was not washed out. Response: The meaning of my statement was that the result of NanoString analysis is based on ‘counts’ not ‘copies’. NanoString assays provide counts that are proportional to underlying copy number. We agree with the Reviewer.
    4. But my main question was that for some miRNAs that you consider in the manuscript, in 10-20% of samples (both cancer and normal) the values are at the level of negative controls, despite the fact that there are no QC flags. This situation would be worth analyzing, since it can significantly affect both the results and the conclusions.
    Response: Background correction has been implemented in data analysis to more precisely delineate which counts are false positive according to the NanoString recommendation. Low expression miRNAs (at the level of negative controls) Non-specific counts have been subtracted from the raw counts to obtain a new estimate of counts above background. Background subtracted values may subsequently be multiplied by normalization scaling factors, thus no samples with QC flags (especially in normalization) were used in the study. Different miRNA expression pattern in samples may be explained by heterogeneity of the miRNA profile. This situation would be worth analyzing in the future. We are very grateful for this suggestion.
    In summary, we would like to thank you once again for such an accurate analysis of our manuscript and all the valuable recommendations.
    Sincerely,

Mariusz Rogucki
Department of Endocrinology, Diabetology and Internal Medicine

Medical University of Bialystok,

15-276 Bialystok, Poland

Reviewer 2 Report

The authors have addressed my concerns in this revised manuscript.

Author Response

Thank you for your comment.

Reviewer 3 Report

In their response, the authors have agreed that their results are only preliminary since they have compared PTC vs normal thyroid gland tissue. The manuscript has been revised in this regard.

The authors have stated in their response that the miRNA expression pattern was different between metastatic and non-metastatic PTC cases. Metastatic PTC cases were excluded based on the following statement:

„Thus, considering additional comorbidities influence miRNA profile, we decided to exclude metastatic patients from the study.”

Metastasis is not considered a comorbidity in case of cancer. Metastatic potential is the fundamental biological characteristic differentiating benign and malignant neoplasms. If the proposed diagnostic tool of the authors is not capable of differentiating metastatic PTC from normal thyroid gland tissue, that should be clearly stated in the manuscript and thoroughly discussed.

Some revision of the manuscript is still warranted. Two possible solutions are the following:

The authors could state that their intention was to see if the miRNA profile of early PTC was different from normal thyroid gland. This could be the argument why they have excluded metastatic cases. However, it should be clearly stated that metastatic cases were excluded, even in the discussion, not just the Methods section.

The authors could report the performance of their test if metastatic PTC cases are included. In this case, further discussion may be necessary, especially, if the performance is reduced. The authors should discuss the potential significance of a different miRNA profile of metastatic vs non-metastatic PTC cases. This may require a substantial number of metastatic PTC cases that the authors may not have included in their project.

Author Response

Dear Reviewer,

Thank you for your constructive comments concerning our manuscript entitled ‘Expression Profile and Diagnostic Significance of MicroRNAs in Papillary Thyroid Cancer’. We have studied your comments carefully and made major correction which we hope to meet your approval.

Our answers to your points are as follows.

  1. Some revision of the manuscript is still warranted. Two possible solutions are the following:
    The authors could state that their intention was to see if the miRNA profile of early PTC was different from normal thyroid gland. This could be the argument why they have excluded metastatic cases. However, it should be clearly stated that metastatic cases were excluded, even in the discussion, not just the Methods section.
    The authors could report the performance of their test if metastatic PTC cases are included. In this case, further discussion may be necessary, especially, if the performance is reduced. The authors should discuss the potential significance of a different miRNA profile of metastatic vs non-metastatic PTC cases. This may require a substantial number of metastatic PTC cases that the authors may not have included in their project.

Response: We agree with the Reviewer. We revised the manuscript and described the aim as the evaluation of miRNA expression patterns in early PTC compared to normal thyroid gland tissue. Moreover, we discussed the metastasis of PTC in the following sentences: ‘Our miRNA model distinguishes between early PTC and normal tissue, thus additional studies including benign tissues should be performed to confirm the diagnostic usefulness of the miRNA panel. What should be also noted is that proposed diagnostic tool is not capable of differentiating metastatic PTC from normal thyroid gland tissue Metastatic PTC were excluded to obtain homogenous PTC cohort. This was also helpful to comprehensively describe tumor-initiating alterations in miRNA expression pattern. Literature data suggest distinct miRNA expression pattern associated with lymph node metastasis in PTC patients. However, the data on diagnostic value of the miRNA profile in differentiation between metastatic PTC and non-metastatic PTC patients is limited.’

In summary, we would like to thank you once again for such an accurate analysis of our manuscript and all the valuable recommendations.

Sincerely,
Mariusz Rogucki
Department of Endocrinology, Diabetology and Internal Medicine Medical University of Bialystok, 15-276 Bialystok, Poland

Round 3

Reviewer 1 Report

I cannot say that I am completely satisfied with the result, but at least progress is visible. In this form, the article can already be published.

There are some minor remarks left. It is necessary in all cases when the p-value is indicated (there are many in Table 2), give the real value, and not only 2 decimal places.

In addition, in the added fragments, not all the sentences are completely clear to me. For example: 

Lines 450-451 "MiRNAs can easily be detected in various types of specimens by methods such as microarrays, nCounter technology, or RT-PCR in peripheral blood."

Perhaps, moderate English editing required.

Author Response

Dear Reviewer,
Thank you for your constructive comments concerning our manuscript entitled ‘Expression 
Profile and Diagnostic Significance of MicroRNAs in Papillary Thyroid Cancer’. We have 
studied your comments carefully and made minor correction which we hope to meet your 
approval. Our answers to your points are as follows:
1. There are some minor remarks left. It is necessary in all cases when the p-value is 
indicated (there are many in Table 2), give the real value, and not only 2 decimal 
places.
Response: p-values were given as suggested.
2. In addition, in the added fragments, not all the sentences are completely clear to me. 
For example: Lines 450-451 "MiRNAs can easily be detected in various types of 
specimens by methods such as microarrays, nCounter technology, or RT-PCR in 
peripheral blood." Perhaps, moderate English editing required.
Response: English editing has been performed.
In summary, we would like to thank you once again for such an accurate analysis of our 
manuscript and all the valuable recommendations.
Sincerely,
Mariusz Rogucki
Department of Endocrinology, Diabetology and Internal Medicine
Medical University of Bialystok,
15-276 Bialystok, Poland